# The evolution of behavioral cues and signaling in displaced communication

**Arthur Bernard**[1]*, **Steffen Wischmann**[1,2], **Dario Floreano**[2], **Laurent Keller**[1,3]

**1** Department of Ecology and Evolution, University of Lausanne, Lausanne, Switzerland, **2** Laboratory of Intelligent Systems, Ecole Polytechnique Fédérale de Lausanne, Lausanne, Switzerland, **3** Social Evolution Unit, Cornuit 8, Chesières, Switzerland

☭ These authors contributed equally to this work.
* arthur.bernard1204@gmail.com

**Data Availability Statement:** The code is available on Github at https://github.com/CAThanatos/RefComm. The evolutionary data was deposited on Zenodo: https://zenodo.org/record/7723119#.ZBDvh3bMIuV.

## Abstract

Displaced communication, whereby individuals communicate regarding a subject that is not immediately present (spatially or temporally), is one of the key features of human language. It also occurs in a few animal species, most notably the honeybee, where the waggle dance is used to communicate the location and quality of a patch of flowers. However, it is difficult to study how it emerged given the paucity of species displaying this capacity and the fact that it often occurs via complex multimodal signals. To address this issue, we developed a novel paradigm in which we conducted experimental evolution with foraging agents endowed with neural networks that regulate their movement and the production of signals. Displaced communication readily evolved but, surprisingly, agents did not use signal amplitude to convey information on food location. Instead, they used signal onset-delay and duration-based mode of communication, which depends on the motion of the agent within a communication area. When agents were experimentally prevented from using these modes of communication, they evolved to use signal amplitude instead. Interestingly, this mode of communication was more efficient and led to higher performance. Subsequent controlled experiments suggested that this more efficient mode of communication failed to evolve because it took more generations to emerge than communication grounded on the onset-delay and length of signaling. These results reveal that displaced communication is likely to initially evolve from non-communicative behavioral cues providing incidental information with evolution later leading to more efficient communication systems through a ritualization process.

## Author summary

The evolution of displaced communication, the process through which individuals share information about a remote object (in space or time), is a key innovation in language. By conducting experimental evolution we found that displaced communication is more likely to leverage and evolve from behavioral cues, such as the agent's movement, rather than from dedicated communication modes, such as the amplitude of emitted signals. This phenomenon is shown to happen because communication via signal amplitude -although

**Funding:** LK received Swiss NSF (https://www.snf.ch/en) grant nr. CR32I3_141063.The funders had no role in study design, data collection and analysis, decision to publish, or preparation of the manuscript.

**Competing interests:** The authors have declared that no competing interests exist.

more efficient- is slower to evolve. The simple behaviors and neural networks of the agents studied here, also suggest that communication may evolve more frequently than expected via ritualization, a process whereby an action or behavior pattern in an animal loses its original function but is retained for its role in display or other social interactions.

## Introduction

The evolution of communication, wherein privately acquired information is transmitted in a social context, still represents a major issue in evolutionary biology [1, 2, 3]. In particular, the origin of displaced communication [4, 5], where individuals communicate on remote or non-visible objects or organisms, is poorly understood. Displaced communication is very common in humans [5] and relatively rare in other organisms. It has also been documented in a few species such as chimpanzees [6, 7], dolphins [8] and parrots [9]. One of the most striking example is the honeybee waggle dance [10], where foragers returning to the hive provide information on the quality and spatial location of foraging sites by modifying the orientation of the dance according to the relative position of the sun to the food source and modulating the length of the waggle proportion according to the distance of the food from the hive [11, 12, 13, 14, 15, 16]. Since historical origins of natural languages cannot be observed directly [17], and because displaced communication often involves complex multimodal signals (e.g., the orientation and the length of the dance in honeybees), studying their origin is challenging [18, 19].

To investigate the evolution of displaced communication, we conducted experimental evolution with simple simulated robots that could make use of a dedicated communication system to provide information on food location [20, 21, 22, 23]. Each experiment replicates the evolution of these artificial organisms from scratch under new environmental conditions. In each experiment, a signal sender and a signal receiver were placed on a one-dimensional circular environment containing a region ("nest") where they could communicate, and five non-over-lapping foraging sites, only one of which contained food at any given time (Fig 1). Agents could freely move clockwise and counterclockwise on the perimeter of this circle by varying their angular velocity. The performance of each pair of sender-receiver agents was evaluated as the proportion of the time spent by the receiver on the foraging site containing food. Importantly, the sender, but not the receiver, could perceive the presence of food when at a site containing food. Communication between the sender and receiver was possible only when they were simultaneously in the communication area in the nest. There, the sender had the possibility to produce a signal whose amplitude could vary continuously and the receiver could potentially use this information to infer which of the foraging sites contained food. The performance of each pair of sender-receiver agents was evaluated in the last 20 time steps (out of 100) of each trial as the proportion of the time spent by the receiver on the foraging site containing food. Experimental evolution was conducted over 25'000 generations in 40 independent populations each containing 1'000 pairs of senders and receivers. Each pair was evaluated during five trials; food was located once at each of the five foraging sites in random order.

The behavior of the agents at each time step was determined by a neural network. The specifications of the agents' neural networks (i.e. the networks' connection weights) were encoded in an artificial genome. The probability of transmission of genomes from one generation to the next was proportional to the performance of the agents. All experiments were initiated with completely naive agents (i.e., with randomly generated genomes that corresponded to randomly wired neural networks) with no information about how to move and identify

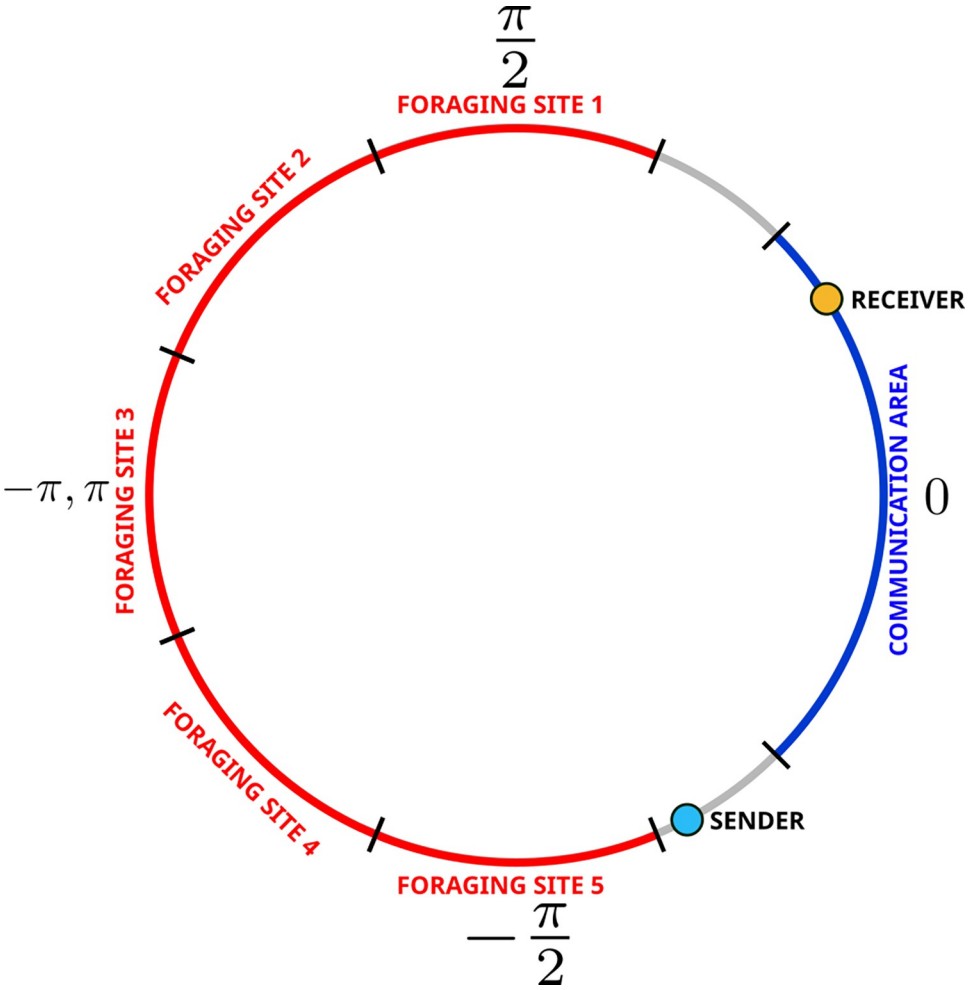

**Fig 1. Illustration of experimental setting.** At each trial, food was randomly located on one of the five foraging sites (marked in red) equally spaced on a one-dimensional circle. Both the sender and the receiver always started a trial at position 0 on the circle located within the communication area, which acted as a "nest" (marked in blue). Agents moved on the border of the circular environment in a direction or the other (clockwise or counterclockwise). See Materials and Methods for a complete description of the experimental setup.

foraging sites or the nest location. Genome mutations occurred with a given probability at each of the 25'000 generations.

## Results

### Role of communication on performance

To determine whether communication evolved and, if so, quantify how it influenced performance, we first conducted experimental evolution under two different treatments. In the communication treatment, the sender could freely signal while in the no-communication treatment we prevented agents from communicating by fixing the signal perceived by the receiver to a constant zero value. Foraging efficiency rapidly increased in both the communication and no-communication treatments (Fig 2), but rose to be over double as high in the communication treatment than in the no-communication treatment (last generation: Mann-Whitney U test, $P < 0.0001$).

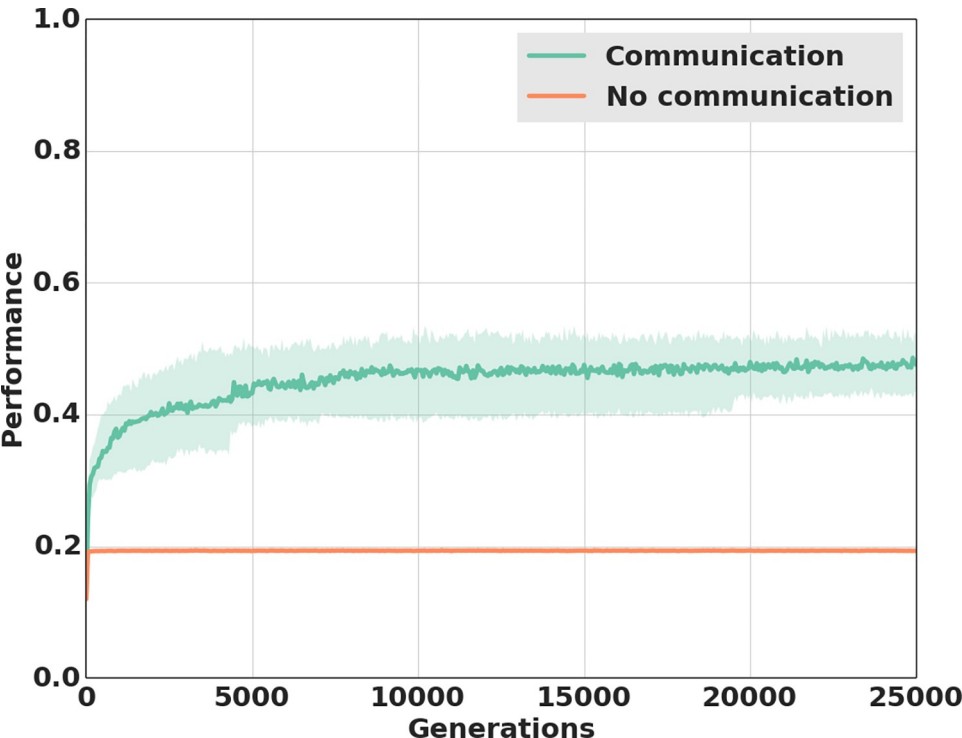

**Fig 2. Performance over generations of evolution depending on communication capabilities.** Performance (i.e., the proportion of time spent by the receiver on the site containing food) over the 25'000 generations of selection when individuals could freely communicate (green line) and when they were prevented from communicating by setting the signal perceived by the receiver to a constant zero value (orange line). Each experimental treatment was replicated over 40 populations. The colored areas represent the first and third quartiles.

Analyses of individual behaviors provided an explanation for the higher performance in the communication treatment compared to the no-communication treatment. In the no-communication treatment, 67% of the receivers evolved the behavioral strategy of going to the same foraging site in each of the 5 trials (Fig 3A). Because food was placed at a different site in each of the 5 trials, these individuals always found food in only one trial, hence leading to a foraging

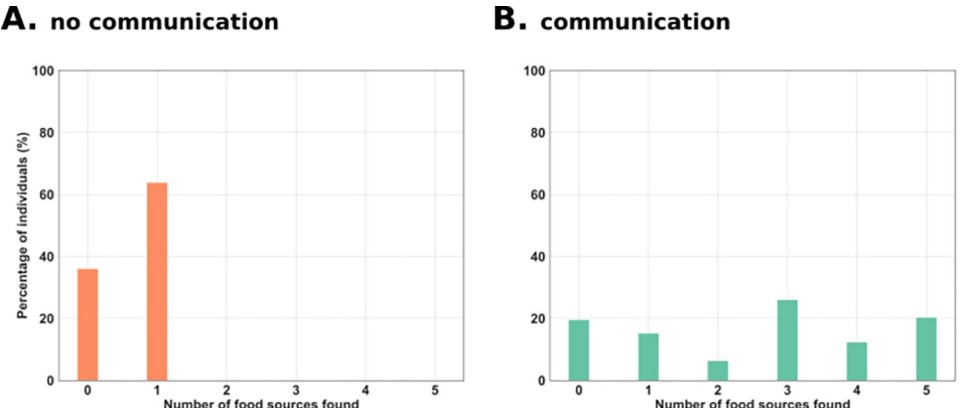

**Fig 3. Repartition of sites found depending on communication capabilities.** Percentage of individuals, which found food 0, 1, 2, 3, 4 or 5 times during the five trials in the (A) no-communication and (B) communication treatments. In the no-communication treatment, receivers perceived a signal with a constant zero value. Receivers were considered to have found food if they spent at least 15 of the last 20 time steps of a trial on the site containing food.

performance close to 0.2 (i.e., the individual spent almost 20% of its lifetime on a foraging site containing food; see S1A Fig and S1 Video for a display of this behavior). The remaining 33% of the agents displayed a different behavioral strategy. They moved slowly throughout the five foraging sites thereby spending about 20% of the time on each of the five foraging sites (see S1B Fig and S2 Video for a display of this behavior). Their performance was therefore close to 0.2 too. Given that both types of agents always spent less than 75% of the time on the foraging site containing food, they were classified as not having found the food (Fig 3A). By contrast, in the communication treatment 68% of the receivers were able to locate food in two or more of the five trials (Fig 3B). This behavior led to a much higher foraging performance than in the no-communication treatment (Fig 2), thus revealing that an effective mode of displaced communication evolved between senders and receivers.

## Mode of communication

Since the neural networks of the senders were designed to allow them to vary the signal amplitude, we expected that senders would signal which site contained food by using this mode of communication as it would have provided a large potential of expressiveness to evolve. Surprisingly, however, there was no consistent difference at the end of the evolutionary experiments (generation 25'000) in signal amplitude depending on the site at which food was located. Overall, the variation of signal amplitude between the five trials was only $0.005 \pm 5.03*10^{-5}$. This unexpected result suggests that senders did not use signal amplitude to provide information on food location. This was confirmed in an additional experiment where we constrained the evolved senders to produce a signal of fixed amplitude, irrespective of the site at which food was located. This manipulation did not lead to a significant reduction in foraging performance (mean performance: $0.471 \pm 0.005$) compared to the treatment where signal amplitude was not constrained (mean performance: $0.472 \pm 0.005$; Mann-Whitney U test $P > 0.4$), confirming that receivers did not use signal amplitude to localize food.

Given that individuals did not transmit information by means of signal amplitude, we hypothesized that they instead used another mode of communicating food location when they were simultaneously in the nest. As a reminder, in the experiments receivers could perceive a signal only when both the sender and receiver were simultaneously in the communication area in the nest. Thus, information on food location could be provided to the receiver either by the delay from the start of the trial to the time when the signal was first perceived by the receiver in the nest (i.e., onset-delay, Fig 4B) or by the amount of time when both the sender and the receiver were simultaneously in the nest (i.e., signal duration, Fig 4C). To test these two hypotheses, we experimentally manipulated both the onset-delay and signal duration of the evolved populations and measured performance during the last generation (see Materials and Methods). These experiments revealed that, depending on historical contingencies, populations evolved to rely on either source of information or both. Shifting the start time of signal production (onset-delay) resulted in a significant decrease in performance in 38 out of the 40 populations (Mann-Whitney U test, $P < 0.01$, Fig 5, orange bar), indicating that the timing of onset-delay was used as a vector of information in most populations (see S2A Fig and S3 Video for a display of this behavior). Constraining the signal duration resulted in a significant decrease in performance in 21 of the 40 populations (Mann-Whitney U test, $P < 0.01$, Fig 5, green bar; see S2B Fig and S4 Video for a display of this behavior). In 20 of the 40 populations, performance was significantly decreased both when the timing of onset-delay of signaling was shifted and when the signaling duration was constrained, indicating that these populations relied on both modes of communication for food location (Mann-Whitney U test, $P < 0.01$, Fig 5, purple bar; see also S3 Fig for detailed analysis of performance in every population).

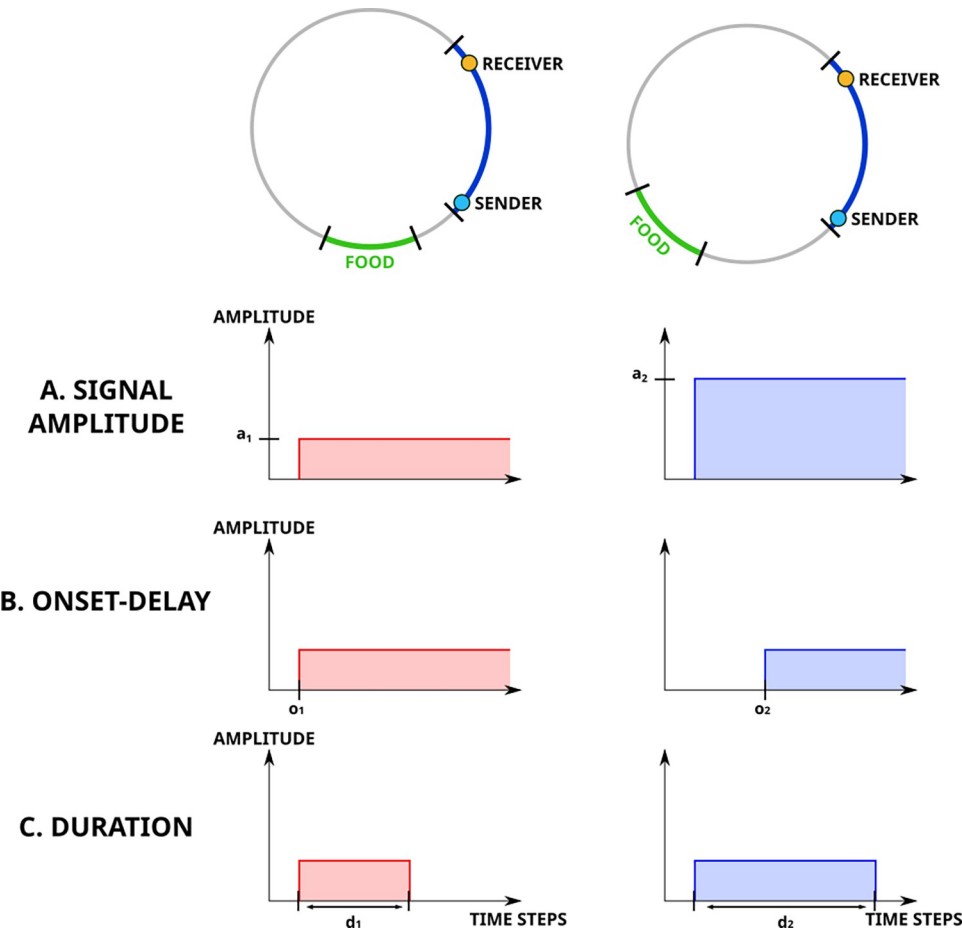

**Fig 4. Illustration of the different modes of communication.** This figure shows how signal reception can be used to provide information on food location. Each column represents a trial where food (in green) is located at a different foraging site (namely foraging site 4 on the left and foraging site 5 on the right). Each rows illustrates how the signal perceived by the receiver inside the communication area changes depending on the mode of communication used by the agents. A) By using signal amplitude, the signal amplitude emitted by the sender is varied to convey information on food location. B) When using onset-delay, the varying delay from the start of the trial to the time when the signal was first perceived by the receiver in the nest provides information on food location. C) Finally when using signal duration, amount of time when both the sender and the receiver were simultaneously in the nest gives information on food location.

The finding that receivers used the timing of onset-delay and duration of signaling as a source of information raises the question of how this evolved. We hypothesized that food location directly influenced the time of arrival of the sender in the communication area and therefore might have affected the timing of onset-delay and/or duration of signaling because senders would arrive earlier and signal longer when food was located at a foraging site close to the nest. Hence, the time of arrival to the nest would have first served as a cue, inadvertently providing information to the receiver about food location. To test this hypothesis, we determined for each population whether there was an association between two evolutionary events: when the timing of signal onset-delay first varied according to which site the food was located and when the average performance first became significantly greater than 0.2 (i.e., the highest performance value achieved by no-communication populations; see Materials and Methods). In all the 40 populations there was a close match between these two values with a performance value significantly greater than 0.2 being reached only 1.4±2.1 generations (Fig 6) after the

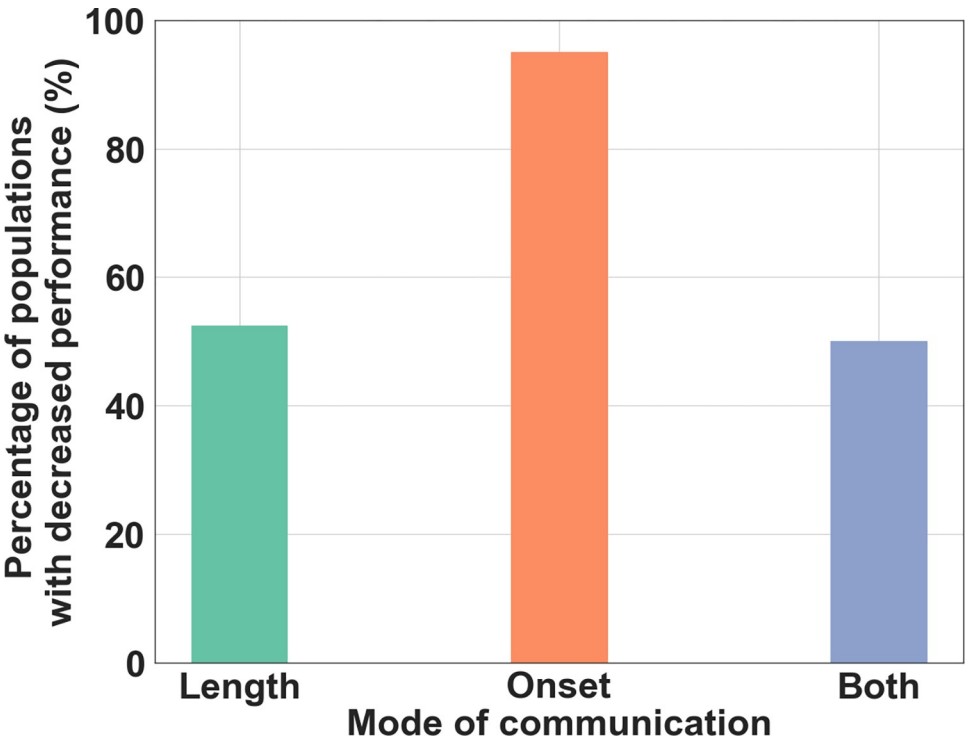

**Fig 5. Percentage of populations with decreased performance depending on communication mode.** Percentage of the populations (n = 40) where performance was significantly (Mann-Whitney U test, $P < 0.01$) decreased by preventing receivers from using information on the timing of signal onset-delay and/or duration. Length (green bar) represents the percentage of populations where performance was decreased when constraining the signal duration. Onset (orange bar) represents the percentage of populations where performance was decreased when shifting signal onset-delay. Both (purple bar) represents the percentage of populations where performance was decreased when altering either the onset-delay or signal duration.

sender changed the timing of signal onset-delay depending on the site at which food was located. Accordingly, there was a strong correlation across populations (Pearson correlation, R = 0.90, P < 0.0001) between the number of generations required for senders to vary in their time of arrival to the nest as a function of food location and the number of generations required for average foraging performance to exceed 0.2.

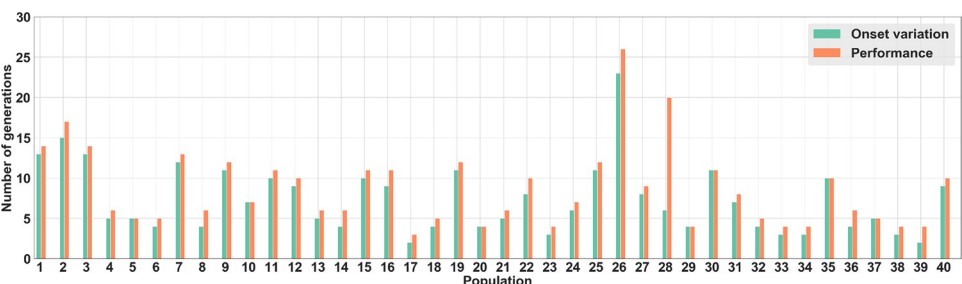

**Fig 6. Number of generations before using signal onset-delay and surpassing performance threshold.** For each of the 40 populations, number of generations required before senders changed signal onset-delay depending on the site at which food was located (in green). Number of generations required for the mean population performance to surpass the maximum performance (i.e., 0.2) in the no-communication populations (in orange).

### Association between the mode of communication and performance

Given the unexpected result that individuals used the onset-delay and duration of signaling in the nest instead of signal amplitude as a mode of communication, we conducted new experiments to investigate whether the agents would evolve the use of signal amplitude when prevented from using variation in signaling onset-delay and duration. This "constrained communication" experiment was performed by forcing senders to always move at a fixed velocity and in the same direction, hence preventing variation in time of arrival to the nest and time spent within the nest.

The elimination of variation in signal onset-delay and signal duration did indeed lead to the evolution of a communication system based on variation of signal amplitude (see S4 Fig and S5 Video for a display of pair of agents using signal amplitude to communicate). In this constrained treatment, the mean variation of signal amplitude between trials was 0.185±0.006, a value significantly greater (Mann-Whitney U test, P <0.0001) than in the unconstrained treatment, where there was almost no variation in signal amplitude between trials (0.005 ±5.03*$10^{-5}$). Importantly, at the end of the experiment, the foraging performance in the constrained communication treatment (0.510±0.008) was also significantly higher than in the unconstrained communication treatment (0.472±0.005; Mann-Whitney U test, P<0.01).

The finding that communication mediated by onset-delay and signaling duration was less efficient than communication mediated by variation in signal amplitude raises the question of why individuals did not use the latter (more efficient) mode of communication in the unconstrained evolutionary experiments. A possible explanation is that a system of communication mediated by signal amplitude is slower to evolve than communication based on signaling onset-delay/duration. To investigate this hypothesis, we compared how the foraging performance evolved over the first 5'000 generations in the constrained and unconstrained treatments (Fig 7). This analysis revealed that unconstrained populations achieved a performance higher than 0.2 (the maximum value that can be reached without communication) faster than constrained populations, whose forgaing performance stayed at 0.2 for almost 800 generations. Overall, unconstrained populations were 42 times faster than the constrained populations to evolve a system of communication corresponding to a foraging performance higher than 0.2 (Mann-Whitney U test, P < 0.0001). Among the 40 constrained populations, the huge variation in the number of generations (between 20 and 5000) required to achieve a foraging performance above 0.2 indicates high variability in the evolutionary time required to discover a communication method based on signal amplitude.

## Discussion

Our analysis showed that experimental evolution with simple artificial agents readily led to the emergence of displaced communication providing information about the location of remote food sources. Surprisingly, although the agents were imbued with a dedicated signaling channel that could vary in amplitude, they did not use signal amplitude as a mode of communication. Instead, they used either the timing of onset-delay and/or duration of signaling in the nest to communicate food location. Importantly, our analysis showed that this outcome was not due to the agents being unable to evolve a system of communication based on signal amplitude. Indeed, when experimentally prevented from using the signal onset-delay and duration, the agents were able to make use of signal amplitude as a mode of communication in all of the 40 evolved populations. These findings are of particular interest because it has been argued that a satisfactory account of the origins of communication with computational models has been hindered by the fact that those models have consistently implemented communication as an exchange of signals over dedicated and functionally isolated channels [24]. Our results

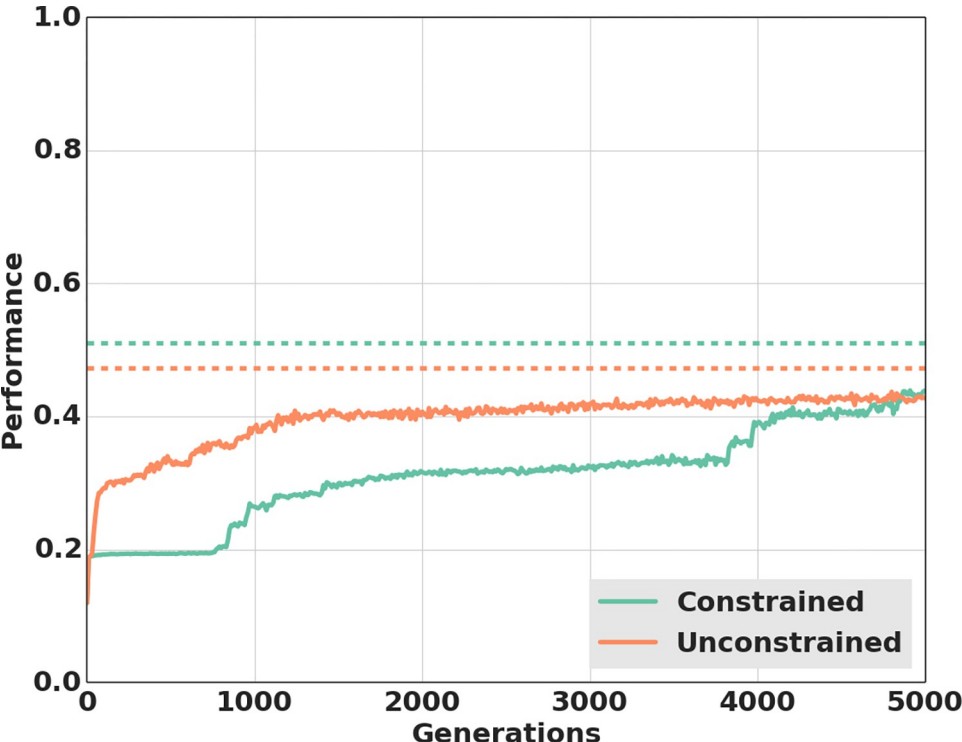

**Fig 7. Average performance in the first 5'000 generations of evolutions depending on treatment.** Average performance in the unconstrained treatment (40 populations) and in the constrained treatment (40 populations) over the first 5'000 generations. In the constrained treatment, senders where forced to move at a fixed velocity, thus preventing the use of signal onset-delay and/or duration by receivers. Dashed lines show the average performance achieved in the two treatments at the end of the experiments (i.e., at generation 25'000).

support the argument that it is possible to evolve communication without dedicated channels and that an understanding of how communication evolves in such situations is of particular interest to our understanding of the evolution of communication in natural systems.

Interestingly, many intraspecific signaling systems in animal groups with evolutionary overlapping interests (i.e., when the fitness of one individual depends, at least in part, on the fitness of the other individual, as in the experiments described here) have been shown to also rely on variation in signaling amplitude or duration (e.g. [25, 26, 27]) or variation in rhythm [28]. This can be understood since such variation readily allows effective communication and because it can quickly evolve, as demonstrated by our experiments.

A surprising result was that communication via signal amplitude was actually more efficient than communication based on signaling onset-delay/duration, raising the question of why agents invariably evolved the less efficient mode of communication in the unconstrained treatment. We hypothesized that this might be due to communication via signal amplitude taking more time to evolve than communication via the timing of signal onset-delay/ duration. Our time-course analyses confirmed this hypothesis, showing that on average it took 42 times more generations for communication to evolve (i.e., for populations to reach a higher performance than that attained by populations where communication of any form was impossible) via signal amplitude than via signal onset-delay/duration. This finding can be explained by the fact that the time taken for agents to return to the nest was rapidly associated with the distance between food and the nest, thereby providing a useful cue to the receiver. By contrast, despite being more efficient, communication via signal amplitude was much slower to evolve—most

likely because it first required that signal amplitude varied non-randomly among senders according to food location before the receivers could evolve an appropriate response. In line with this view of a slow stochastic process being required for reaching a difficult evolutionary target, we showed that there was a large variation across populations in the number of generations required to surpass a performance score of 0.2 (the performance hallmark of some form of communication) when agents could not use signal onset-delay/duration to communicate and had to rely on signal amplitude.

Interestingly, our analysis revealed that once the populations had evolved a mode of communication based on the timing of signal onset-delay and/or signal duration, none of them were able to switch to the more efficient system of communication via signal amplitude. A likely reason for this is that switching from one system of communication to the other would require passing through a valley of lower performance values [29] where each population would have to abandon their original mode of communication to develop the other. This problem is likely to be particularly acute in the case of communication systems because changes in either the signaling or response strategy would destroy the communication system that is already in place and result in performance decrease [30, 31]. This may account for some of the differences in signaling observed between closely related species and isolated populations of a given species. For example, Anolis lizards originating from different evolutionary ancestors have evolved different signaling systems in response to similar selective pressures [32]. It is also possible that a new mode of communication could evolve and coexist with the original mode of communication but this would probably require that it does not interfere with it. This is because once a mode of communication has evolved, individuals changing their signaling or response strategy are likely to have lower performance. But new modes of communication may evolve when this does not lead to a disruption of the original mode of communication.

This study also supports the hypothesis that communication may often evolve via ritualization, a process whereby an action or behavior pattern in an animal loses its original function to serve as a mode of display or other role in social interactions [33]. The path to communication revealed by our study is typical of such a process [2, 34]. The variations of motor actions initially acted as non-selected cues (i.e., timing of return to the nest and time spent within the nest) that elicited an adaptive reaction in the receiver. Through selection, these cues then became full-fledged signals that provided reliable information about food location. Similarly, ritualization has been proposed as a route towards the evolution of the waggle dance in honeybees [35, 36]. Because communication based on the timing of signal onset-delay/duration is built from an existing behavior, its evolution was faster than that of communication based on signal amplitude which required the coordinated evolution of signal and response (i.e., coordination between senders and receivers). As a result, communication by signal amplitude never evolved when the agents could use the timing of signal onset-delay and/or signal duration in the unconstrained treatment. In conclusion, our study reveals that ritualization may play a more pervasive role than realized, in particular for the emergence of displaced communication.

## Materials and methods

### Experimental setup

The environment was a one-dimensional circle containing a nest where individuals could communicate with each other and five non-overlapping foraging sites at fixed positions: $\pi/2$, $3\pi/4$, $\pi$, $-3\pi/4$ and $-\pi/2$ (Fig 1). Each foraging site's length was $\pi/4$. Each trial was always conducted with a *sender* and a *receiver*, both of which were initially located at position 0 on the circle. Each pair was evaluated over five trials, with food being located at each foraging site once

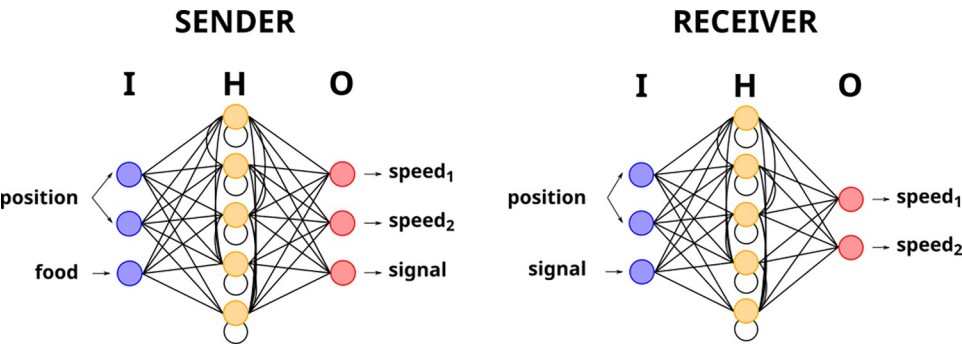

**Fig 8. Illustration of agent neural controllers.** The agents' controllers were comprised of two continuous-time, recurrent neural networks (CTRNN) with a fully connected hidden layer of five neurons. The input and output units of the agents differed to match the specific sensory and communication apparatus of signalers and receivers. Connections between neurons are represented by black lines and input, hidden and output layers are respectively designated as *I*, *H* and *O*.

and changed position at each trial in random order. The performance of each pair of sender-receiver agents was proportional to the number of time steps spent by the receiver on the foraging site containing food during the last 20 time steps (out of 100) of each of the five trials. Starting from 0, performance was increased by 0.01 for each time step spent at the food location and thus the maximum performance achievable after five trials was equal to 1 [0.01*20*5].

The agent could move around the circle in either direction (i.e., clockwise or counterclockwise) and freely vary their speed (i.e., angular velocity) from zero to a maximum of $\pi/9$. Agents could pass each other without collision. In addition, the sender was equipped with a floor sensor for food detection that would switch from 0 to 1 if food was present at a given foraging site. The sender was also equipped with a signalling output unit and at each time step could vary its amplitude in the continuous range between 0 and 1, which the receiver could perceive only when both agents were in the nest. The nest was centered on 0 and was $\pi/2$ wide (i.e., extending from $-\pi/4$ to $\pi/4$). Outside of this area, the signal perceived by the receiver was equal to zero, independent of the amplitude of the signal sent by the signaling agent.

Each agent was controlled by an individual neural network that, given a set of inputs representing the sensory information of the agent, computed its desired behavior at each time step. In particular, agents were endowed with continuous-time recurrent neural networks (CTRNN) [37] (Fig 8), which have been frequently applied in evolutionary robotics [24, 38]. In contrast to feedforward neural networks, CTRNN form a directed acyclic graph which allows them to store an internal state akin to a memory. This enables the network to display dynamic temporal behavior and act on inputs removed in time, a feature especially suited for the evolution of displaced communication. To compute the activation of its neurons, a CTRNN uses an ordinary differential. Activation of a given neuron i at time step t is computed following the Euler step as follows:

$$y_{it} = \sigma(s_{it} + \theta_i),$$

where σ is the sigmoid activation function, $\theta_i$ the bias term for neuron i and $s_{it}$ the state of neuron i at time step t. $s_{it}$ is calculated as:

$$s_{it} = \alpha * \tau_i * \left(\sum_j W_{ji} * y_{jt} - s_{i(t-1)}\right),$$

where $\alpha$ is the step size (0.1 in our experiments), $\tau_i$ is the time constant for neuron i, $W_{ji}$ the

connection weight from neuron j to neuron i, $y_{jt}$ the activation of neuron j at time step $t$ and $s_{i(t-1)}$ the stage of neuron i at time step $t-1$.

Both senders and receivers possessed two inputs indicating their own location (expressed as sine and cosine values). The sender had an additional input providing information on whether the foraging site on which it was contained food (1 if food was present, 0 otherwise) and the receiver had an additional input corresponding to the perceived signal amplitude. Each network included five hidden neurons with recurrent connections and two output neurons. The two output neurons controlled the speed and direction of the agent. Speed was computed as the absolute value of the difference between these two outputs and direction as the sign of this difference. In addition, the sender had a supplementary output encoding the signal amplitude. Neural networks were allowed to evolve via the connection weights between neurons. Each synaptic connection was encoded in a single gene whose real value was in the [0, 1] range. It was then mapped linearly in the [–4, 4] range to be used as a connection weight. The neuron's integration time constant $\tau$ and bias term $\theta$ were encoded in genes whose values were linearly mapped in the [0.1, 1.0] range and the [–2, 2] range respectively. This amounted to a total genome size of 77 ($3_{input}*5_{hidden} + 5_{hidden}*5_{hidden} + 5_{hidden}*3_{output} + 11_{bias} + 11_{timeconstant}$) values for the sender and 70 ($3_{input}*5_{hidden} + 5_{hidden}*5_{hidden} + 5_{hidden}*2_{output} + 10_{bias} + 10_{timeconstant}$) values for the receiver.

## Artificial evolution

Each of the 40 independent populations comprised 1'000 pairs of senders and receivers. At generation 0, each gene in the genome was initialized with a random value uniformly sampled from the [0, 1] range. Each sender was randomly paired with a receiver from the same population. The performance of this pair was evaluated across five trials; in each trial, food was randomly positioned at one of five different foraging sites. For a given trial $T$, performance $p_T$ was calculated as follows:

$$p_T = \sum\nolimits_{t=81}^{100} \mathbb{I}_{Tt}*0.01,$$

where $\mathbb{I}_{Tt}$ is a function that returns 1 if the receiver is on food location at step $t$ and 0 otherwise. As such, the maximum performance achievable per trial $T$ was 0.2 and 1 for the five trials.

After the performance of every pair had been evaluated, tournament selection [39] was separately applied to each group (tournament size = 10) to identify the 1'000 pairs of senders receivers selected to produce the next generation. Each gene of a selected genome was mutated with a mutation rate $\mu_N$ which depended on the genome size, as follows:

$$\mu_N = \frac{\mu}{G},$$

where $\mu$ is the baseline mutation rate, whose value is set to 0.5, and $G$ is the genome size. Thus, each gene had a mutation probability of $7.1*10^{-3}$ for a sender's genome and $6.5*10^{-3}$ for a receiver's genome. This means that, while genomes were of different length, the expected number of mutations for both genomes each generation should be the same. For each mutated gene, we replaced its value by a random value sampled from a normal distribution. All populations evolved for 25'000 generations.

## Analysis of communication strategies

To assess individuals' performance and strategies, we determined the number of trials in which each receiver successfully found the food location. We considered that food was

successfully found when receivers spent at least 15 of the last 20 timesteps of a trial on the foraging site containing the food (corresponding to a performance of at least 0.15 for a given trial).

To study how signal amplitude varied according to food location, we considered only the trials in which the receiver found the food location (as per the previous definition). For a given sender, we measured the difference in signal amplitude at the same time step between the five different trials. Mean signal variation $\acute{S}_i$ for each individual i was thus calculated as follows:

$$\acute{S}_i = \frac{1}{100}\sum\nolimits_{u=1}^{100}\frac{2}{|T|(|T|-1)}\sum\nolimits_{t_1 \in T}\sum\nolimits_{t_2 \in T, t_2 \neq t_1}|s_{ut_1} - s_{ut_2}|,$$

where T was a set representing the 5 different trials (T = {1, 2, 3, 4, 5}, cardinality $|T| = 5$), $t_1$ and $t_2$ two different trials, $u$ the time step in the range [1, 100] and $s_{ut_n}$ is the signal amplitude emitted by the sender at time step $u$ in trial $t_n$.

To study whether receivers relied on signal amplitude, we fed the receivers' sensors with a signal that did not change with the location of food (i.e., did not change between trials). Specifically, at each time step, the receiver was given a signal whose amplitude was equal, for a given time step, to the average signal amplitude emitted by the sender over the five trials at this time step. As such, the signal amplitude $a_u$ received at time step u was:

$$a_u = \frac{1}{|T|}\sum\nolimits_{t \in T}s_{ut},$$

where $T$, $t$, $u$ and $s_{ut}$ were the same variables as defined in the previous equation.

We identified the first generation where there was a difference in onset-delay between the 5 trials for each sender (i.e., when communication onset-delay was different depending on the foraging site at which food was located) to determine when senders began to actively communicate food location.

To determine whether agents relied on the onset-delay of the signal we tested whether performance was affected when changing the onset-delay of signaling. To that end, the onset-delay of the signal was shifted (in time steps) by a value equal to the average difference of onset-delay value between trials. Receivers would thus perceive it earlier or later than it was actually emitted by the sender. To determine whether agents relied on the signal duration, we forced senders to stay in the communication area once they had entered it. In both cases, the average performance was then compared with control trials.

## Supporting information

**S1 Fig. Behaviors of pairs of sender and receiver in the no communication treatment.** Behaviors of best performing pairs of sender and receiver in the no communication treatment for 3 given trials (i.e. 3 different food locations). Each column corresponds to a different trial. The figures show the position of the sender (resp. receiver) in red (resp. blue) at each of the 100 time steps of the trials. The position on the circle is indicated as the angular position in range [-π, π]. The communication area is indicated in blue and the foraging site containing food in red. The behaviors of two different pairs of sender and receiver are shown here. In (A), the receiver goes to the same foraging site at each trial (i.e. the foraging site located at π/2) while in (B), the receiver moves through every foraging site during the last 20 steps of simulation.
(TIF)

**S2 Fig. Behaviors of pairs of sender and receiver in the unconstrained treatment.** Behaviors of best performing pairs of sender and receiver in the unconstrained treatment for 3 given trials (i.e. 3 different food locations). Each column corresponds to a different trial. The top 3 figures show the position of the sender (resp. receiver) in red (resp. blue) at each of the 100 time steps of the trials. The position on the circle is indicated as the angular position in range [-π, π]. Communication area is indicated in blue and the foraging site containing food in red. The bottom 3 figures display the signal amplitude perceived by the receiver while in the communication area (blue area). The behaviors of two different pairs of sender and receiver are shown here. In (A), the pair uses onset-delay as the way to communicate while in (B) they use length of signaling to transmit information.
(TIF)

**S3 Fig. Performance difference for every population depending on communication constraints.** Performance difference of every population line when (A) communication length was constrained and (B) communication onset was constrained.
(TIF)

**S4 Fig. Behavior of a pair of sender and receiver in the constrained treatment.** Behavior of a best performing pairs of sender and receiver in the treatment where sender velocity was constrained for 3 given trials (i.e. 3 different food locations). Each column corresponds to a different trial. The top 3 figures show the position of the sender (resp. receiver) in red (resp. blue) at each of the 100 time steps of the trials. The position on the circle is indicated as the angular position in range [-π, π]. Communication area is indicated in blue and the foraging site containing food in red. The bottom 3 figures display the signal amplitude perceived by the receiver while in the communication area (blue area).
(TIF)

**S1 Video. Behavior of a pair of sender and receiver in the no communication treatment with a single target behavior.** Behaviors of a best performing pair of sender and receiver in the no communication treatment for 3 given trials (i.e. 3 different food locations). The sender (resp. receiver) is drawn as a red (resp. blue) dot. Communication area is drawn in blue and the foraging target containing food in red. The receiver exhibits a behavior where it goes to the same foraging site at each trial (i.e. the foraging site located at π/2).
(MP4)

**S2 Video. Behavior of a pair of sender and receiver in the no communication treatment with a multiple targets behavior.** Behaviors of a best performing pair of sender and receiver in the no communication treatment for 3 given trials (i.e. 3 different food locations). The sender (resp. receiver) is drawn as a red (resp. blue) dot. Communication area is drawn in blue and the foraging target containing food in red. The receiver exhibits a behavior where it moves through every foraging site during the last 20 steps of simulation.
(MP4)

**S3 Video. Behavior of a pair of sender and receiver in the unconstrained treatment using onset-delay.** Behavior of a best performing pair of sender and receiver in the unconstrained treatment for 3 given trials (i.e. 3 different food locations). The sender (resp. receiver) is drawn as a red (resp. blue) dot. Communication area is drawn in blue and the foraging target containing food in red. The pair uses onset-delay as the way to communicate.
(MP4)

**S4 Video. Behavior of a pair of sender and receiver in the unconstrained treatment using signal duration.** Behavior of a best performing pair of sender and receiver in the

unconstrained treatment for 3 given trials (i.e. 3 different food locations). The sender (resp. receiver) is drawn as a red (resp. blue) dot. Communication area is drawn in blue and the foraging target containing food in red. The pair uses length of signaling to transmit information. (MP4)

**S5 Video. Behavior of a pair of sender and receiver in the constrained treatment.** Behavior of a best performing pair of sender and receiver in the treatment where sender velocity was constrained for 3 given trials (i.e. 3 different food locations). The sender (resp. receiver) is drawn as a red (resp. blue) dot. Communication area is drawn in blue and the foraging target containing food in red. The pair uses signal amplitude to transmit information. (MP4)

## Acknowledgments

The authors thank Kuniaki Noda, Peter Dürr, and Sara Mitri for conceiving and contributing to preliminary experiments that lead to this study and Tom Kay for very useful comments.

## Author Contributions

**Conceptualization:** Steffen Wischmann, Dario Floreano, Laurent Keller.

**Formal analysis:** Arthur Bernard, Steffen Wischmann, Dario Floreano, Laurent Keller.

**Investigation:** Arthur Bernard.

**Supervision:** Dario Floreano, Laurent Keller.

**Writing – original draft:** Arthur Bernard, Dario Floreano, Laurent Keller.

**Writing – review & editing:** Arthur Bernard, Dario Floreano, Laurent Keller.

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
