## [Decision Letter · Decision Letter 0]

31 Oct 2022

Dear Dr. Bernard,

Thank you very much for submitting your manuscript "The evolution of behavioral cues and signaling in displaced communication" for consideration at PLOS Computational Biology.

As with all papers reviewed by the journal, your manuscript was reviewed by members of the editorial board and by several independent reviewers. In light of the reviews (below this email), we would like to invite the resubmission of a significantly-revised version that takes into account the reviewers' comments.

In particular we note that the reviewers find the topic and methods of interest but feel there are several points where there is insufficient explanation or justification of choices made in the modelling, leading to over-general claims about the results. Also, both mention the need for the figures to be more self-explanatory.

We cannot make any decision about publication until we have seen the revised manuscript and your response to the reviewers' comments. Your revised manuscript is also likely to be sent to reviewers for further evaluation.

Sincerely,

Barbara Webb

Academic Editor

PLOS Computational Biology

James O'Dwyer

Section Editor

PLOS Computational Biology

Reviewer's Responses to Questions

**Comments to the Authors:**

Reviewer #1: Abstract

lines 15-17

it is said displaced communication is "one of the key innovations of human language", yet also stated it occurs in a few animal species. Perhaps it should be called a "feature" instead of an innovation (or qualified). On the other hand, it's "just" the abstract, so perhaps this is ok.

Introduction

line 52 - "defining feature of human language" -> I find that statements of this sort are either redudant or, undesirably, conceptually problematic. Is human language without displaced communication not human language anymore? Is it a claim that human language - whatever the authors' definition - would not have evolved without that feature? Especially taking into account that some version of it is present in non-human primates... not really a criticism, more of a well-meaning challenge to the authors

I I fully agree with the authors' conclusions that latching onto an pre-existing behavior explains why populations do not use amplitude as cues (or, to generalize, why they do not use new behaviors as cues when completely unconstrained). But this seems to yield a claim that whatever is already there is co-opted if informative enough (vs. some other more (or equally) informative but unprecedented behavior.) While this makes sense all else being equal, I'm not entirely sure this is the case. Is there a (theoretical or practical) performance treshold above which popoulations would adopt "amplitude"? Would perhaps a performance level that makes up for the "valley of low performance values" make populations more likely to switch? In other words, what gets highly efficient, not obvious communication strategies off the ground?

Another challenge is generalizing this to displaced communication as a feature of human language, in which (in the overwhelming majority of cases) there is no deterministic relationship between what is being communicated and concurrent physical cues. Perhaps the claims here should be more circumscribed to cases such as the one covered by the experiments: "remote/obscured object of interest", and not as general as "subject of communication is remote is space and/or time", which seems fairly trickier.

overall, I find this paper very good and clever, and definitely worthy of publication in this journal. its title and some of the claims might be too ambitious, as pointed out earlier, but the work contained in the paper is at the forefront of how evolutionary robotics & agent-based modelling can tell us a lot about communication (and indeed ultimately language, though the picture is of course extremely complex).

The methods seem adequate and more advanced than those found in papers with a similar scope. it would be interesting to see a leaning into population dynamics in the future.

A minor revision qualifying the claims made about human language and displaced communication at large with make the paper stronger and less open to criticism which would obscure these strengths

suggestion on figures: I would add a bit more information to the legends, namely what is being constrained (for example, onset delay/duraton in Fig 6). It makes sense when reading the paper as it is, but I believe that being more informative will help readers looking for a quick reference

I would also recommend being precise about experiments/figures when referring to them (e.g. lines 295-296: "in the experiments where agents were constrained to use signal amplitude there there was [...]").

Reviewer #2: General remarks:

- At the end of the introduction, it is not clear to me what the receiver agent does with the information received from the sender (the information seems to be missing), which is problematic before going to the results section.

- How are the main claims (e.g. signal amplitude vs others) influenced by the neural network type used? This is an important point which should be clearly discussed and compared to what was done in other studies. Why CTRNNs? and not usual RNN, GRU or LSTM?

- How the hyperparameter were chosen? (mutation rate, number of agents, population size, ...). Are they standard values used in other studies?

- Figures could be more informative and more Figures should be provided, in particular explanatory schemas: different types of communication, example of the steps of how agents behave (first sender look at where is the food before locating the receiver, etc.), which operations done on genes, etc.).

- In general, there is a lack summarization of experimental design and several methodological details are missing (CTRNN is mentioned once and no details are provided). This does not able to reader to properly evaluate if the results are interesting. More general information should be provided, including for people outside the field of communication evolution with NN-based agents.

Small Remarks:

- This does not correspond to figure 1: (line 78-79) "The specifications of the agents' neural networks were encoded in an artificial genome (Fig 1). "

- Some sentences are not undertandable: 156-157: "... provided by variation in when the signal was ...""

- The GitHub is provided, which is good, but the readme is kind of short. The authors should provide more details on how to use their code, and in particular how to reproduce their figures and provide the random seeds used.

- Evolutionary data link is provided, but it seems that the corresponding DOI does not exist: https://doi.org/10.5061/dryad.f4qrfj6zt

**Have the authors made all data and (if applicable) computational code underlying the findings in their manuscript fully available?**

Reviewer #1: Yes

Reviewer #2: Yes

PLOS authors have the option to publish the peer review history of their article (what does this mean?). If published, this will include your full peer review and any attached files.

Reviewer #1: No

Reviewer #2: No
---

## [Decision Letter · Decision Letter 1]

15 Feb 2023

Dear Dr. Bernard,

Thank you very much for re submitting your manuscript "The evolution of behavioral cues and signaling in displaced communication" for consideration at PLOS Computational Biology. We apologise for the delay in processing this revision as we were awaiting comments on the revised paper from the second of the original reviewers, but unfortunately these have not been forthcoming. As a consequence, and to avoid further delay, as editor I am providing some specific recommendations for minor revision to improve the clarity of the paper:

1) In the main paper text, it is easy for the reader to miss the crucial specification (line 68) that the agents are moving in a *one dimensional* circular environment, i.e. only along the perimeter of the 'arena' illustrated in figure 1. Nowhere else in the subsequent description or caption to figure 1 (not until the detailed methods, or supplementary videos) is it made clear that the agents move by varying their angular velocity (speed and direction). Consequently it is hard to follow what behaviour is exhibited by the agents or why, for example, communication onset or duration can be constrained by fixing the sender agent's velocity. Please add a few explanatory phrases about the agent's movement to the main text introducing the environment and to the caption of figure 1.

2) The definition of 'onset-delay' is confusingly phrased (line 164, caption to figure 4) as "the varying delay between the time when the signal was first perceived by the receiver in the nest and the start of the trial". This implies 'first perceived' happens before 'start of trial'. Please revise, where-ever it occurs to "the varying delay from the start of the trial to the time when the signal was first perceived by the receiver in the nest".

3) Line 333, it is not clear why "changes in either the signaling or response strategy would [necessarily] destroy the communication system that is already in place". Indeed it seems very plausible that the existence of one strategy could bootstrap the development of another, more efficient strategy, with both co-existing until the more efficient one takes over. Possibly this did not happen in the current simulation because the potential increase in performance (from 0.47 to 0.51) is relatively small, so much longer evolutionary time would be required to discover it. Please revise the discussion to address this point.

Based on the positive comments from reviewer 1, we are likely to accept this manuscript for publication, providing that you modify the manuscript according to these recommendations.

Sincerely,

Barbara Webb

Academic Editor

PLOS Computational Biology

James O'Dwyer

Section Editor

PLOS Computational Biology

Reviewer's Responses to Questions

**Comments to the Authors:**

Reviewer #1: The authors have diligently addressed all comments made by me and another review.

I have no further comments to make, and support the publication of this paper.

**Have the authors made all data and (if applicable) computational code underlying the findings in their manuscript fully available?**

Reviewer #1: Yes

PLOS authors have the option to publish the peer review history of their article (what does this mean?). If published, this will include your full peer review and any attached files.

Reviewer #1: No

Figure Files:

Data Requirements:

Reproducibility:

References:

---

## [Editor Report · Decision Letter 2]

7 Mar 2023

Dear Dr. Bernard,

We are pleased to inform you that your manuscript 'The evolution of behavioral cues and signaling in displaced communication' has been provisionally accepted for publication in PLOS Computational Biology.

Best regards,

Barbara Webb

Academic Editor

PLOS Computational Biology

James O'Dwyer

Section Editor

PLOS Computational Biology

---

## [Editor Report · Acceptance letter]

22 Mar 2023

PCOMPBIOL-D-22-01238R2 

The evolution of behavioral cues and signaling in displaced communication

Dear Dr Bernard,

I am pleased to inform you that your manuscript has been formally accepted for publication in PLOS Computational Biology. Your manuscript is now with our production department and you will be notified of the publication date in due course.

With kind regards,

Zsofia Freund
